# Illegal Use of Testosterone and Other Anabolic–Androgenic Steroids in the Population of Amateur Athletes in Wrocław, Poland—An Unfavorable Lifestyle Trend in the Population of Men of Reproductive Age

**DOI:** 10.3390/jcm13133719

**Published:** 2024-06-26

**Authors:** Monika Skrzypiec-Spring, Andrzej Pokrywka, Wojciech Bombała, Daria Berezovska, Julia Rozmus, Kinga Brawańska, Konrad Nowicki, Gina Abu Faraj, Michał Rynkowski, Adam Szeląg

**Affiliations:** 1Department of Pharmacology, Wroclaw Medical University, Mikulicza-Radeckiego 2, 50-345 Wroclaw, Poland; julia.rozmus.student@gmail.com (J.R.); kinbraw2@gmail.com (K.B.); konradnow@int.pl (K.N.); abufarajgina4@gmail.com (G.A.F.); adam.szelag@umw.edu.pl (A.S.); 2Chair and Department of Biochemistry and Pharmacogenomics, Medical University of Warsaw, Banacha 1, 02-097 Warsaw, Poland; andrzej.pokrywka@wum.edu.pl (A.P.); daria.berezovska@wum.edu.pl (D.B.); 3Polish Anti-Doping Agency, Fabryczna 5A, 00-446 Warsaw, Poland; michal.rynkowski@antydoping.pl; 4Statistical Analysis Centre, Wroclaw Medical University, K. Marcinkowskiego 2-6, 50-368 Wroclaw, Poland; wojciech.bombala@umw.edu.pl

**Keywords:** reproductive health, infertility, lifestyle, testosterone, anabolic–androgenic steroids

## Abstract

**Background:** One factor that may negatively impact male reproductive health is the illegal use of testosterone and anabolic–androgenic steroids. This study aimed to evaluate the prevalence of testosterone use in recreational athletes, as well as factors associated with its use, and to determine the profile of a person using testosterone. **Methods:** A cross-sectional analysis of data from an anonymous, online questionnaire of men recruited from gyms, randomly selected in Wrocław, Poland, has been performed. The minimal sample size was evaluated with the univariate logistic regression model. The association between testosterone use and other factors was also evaluated with the univariate logistic regression model. **Results:** A total of 35% of respondents used testosterone. The main purposes of testosterone use were the improvement of training effects and the improvement of body shape. The respondents most likely to use testosterone and other anabolic–androgenic steroids were men aged 26–35, whose earnings were at the level of the middle class or higher, who were married, had children, had training experience of at least 6 months, exercised at least once a week, took part in weightlifting competitions, were managers in a corporation or enterprise, or were self-employed. Most of the people using testosterone had self-treated side effects. **Conclusions:** The profile of the person most likely to use testosterone corresponds to the characteristics of men in optimal socio-demographic conditions for reproduction. These results indicate that this is a significant social problem that may impact male reproductive health.

## 1. Introduction

Over the last 40 years, sperm counts have declined worldwide, and semen quality has deteriorated, with 1 in 20 men now experiencing fertility problems [1]. It is estimated that infertility caused by males ranges from 20% to 70%, and the percentage of infertile men ranges from 2.5% to 12%. According to various sources, male infertility rates in Central and Eastern Europe range from 8 to 12%, being, next to those in Africa, among the highest in the world [2].

Many different factors related to the current lifestyle, such as mental stress, obesity or excessive care for a muscular figure, overwork, and the use of stimulants, drugs, and other substances, as well as environmental factors, such as air pollution and heavy metals, may potentially affect men’s reproductive health. One of the most important factors negatively affecting male fertility is the medically unjustified or illegal use of testosterone and anabolic–androgenic steroids.

Testosterone use by men has increased markedly, due to numerous factors, including the increased awareness of androgen deficiency syndromes and the growing off-label use, as well as illegal performance enhancement in elite athletes and amateurs training for body and strength sports to improve their body shape and strength, or as a revitalizing substance in corporate employees seeking to increase their energy, motivation, or vitality, despite no proven organic androgen deficiency. In the United States, there was a significant increase in testosterone prescriptions from 2016 to 2019 [3].

The prevalence of illegal testosterone use is not well documented. Most of the studies have analyzed the prevalence of anabolic steroid or hormone use, but not the use of testosterone itself. This is probably because illegal androgen cycles contain multiple agents. In addition, many other performance- and image-enhancing drugs are used concurrently, including growth hormones, thyroid hormones, and insulin [4,5]. McCabe et al. have assessed the prevalence of the non-medical use of anabolic steroids among U.S. college students [6]. The data were collected through self-administered mail surveys from 15,282, 14,428, 13,953, and 10,904 randomly selected college students at the same 119 colleges in 1993, 1997, 1999, and 2001, respectivley. The authors demonstrated that the prevalence of lifetime, past-year, and past-month anabolic–androgenic steroid use was 1% or less of the examined students, and there were no significant differences between 1993 and 2001. A meta-analysis of 271 articles on the prevalence of androgen abuse by Sagoe et al. revealed that lifetime androgen use averaged at 6.4% for men and 1.6% for women and was higher in recreational sports (18.4%), athletes (13.4%), prisoners (12.4%), drug users (8.0%), and high school students (2.3%) in comparison to non-athletes (1.0%) [7]. The prevalence of self-reported lifetime use of androgens in the United States is 6.6%; 2.8% in South Africa [8,9]; and 3.3% in Australian secondary school students [10]. More recently, Montuori et al. conducted a cross-sectional survey of data from an anonymous questionnaire of 107 bodybuilders recruited from 30 gyms randomly selected in the Naples metropolitan area. The authors showed that 35.51% of the respondents declared the illicit use of hormones, but without indicating the specific substances. Of these, 71.05% were men and 28.05% were women [11]. According to the authors of the cohort study of 100 anabolic–androgenic steroid abusers without health problems (HAARLEM study), testosterone was used by 96% of the participants; therefore, we can speculate that the prevalence of testosterone use should be similar to that of anabolic steroids or hormones [4]. The data cited indicate a different trend in the use of testosterone and anabolic–androgenic steroids and point out that this may constitute a significant social problem, especially in the context of declining male fertility.

Androgens can be defined as substances capable of developing and maintaining male characteristics in reproductive tissues, as well as contributing to the anabolic state of somatic tissues [12,13]. Pharmacologically, androgens may be defined by their binding and activation of the androgen receptor [12]. The principal natural androgen in the circulation of mature males is testosterone and its potent metabolite, dihydrotestosterone (DHT). The pivotal role of T in male sexual development and functioning begins in the uterus and continues through infancy, adolescence, and adulthood. In the first trimester of intrauterine life, it is involved in the differentiation of male genital organs. Recent evidence also indicates that testosterone, along with other androgens, is also involved in differentiating sexual dimorphism in certain areas of the brain, such as the amygdala. It influences growth and virilization during infancy, stimulates the development of secondary sexual characteristics and linear growth acceleration, and exerts a beneficial effect on muscle mass and bone mass growth, as well as alterations of body composition during puberty. After puberty, it maintains virilization [12,13]. Testosterone influences the mood, behavior, self-image, and quality of life of men of all ages [14]. When physiological doses of testosterone are used in appropriate, well-defined indications, serious side effects are rare [12]. However, supraphysiological or massive amounts of androgens and non-physiological androgen use in women or children cause androgenic side effects [15].

The main side effects of androgen abuse include the suppression of reproductive functions, with the depression of spermatogenesis and fertility [12,13]. This is related to the blocking of gonadotropin secretion by the pituitary gland by exogenously administered testosterone, and due to the lack of secondary inhibition of spermatogenesis. The inhibition of spermatogenesis is usually reversible and returns to normal sometime after the discontinuation of anabolic–androgenic steroids. Nevertheless, it has been shown that the use of anabolic–androgenic steroids may cause structural changes in the testicles. Experimental animal models have shown that the use of anabolic–androgenic steroids results in morphological changes in the Leydig cells [16]. Additionally, although the number of Leydig cells increases after a period of discontinuation of anabolic–androgenic steroids, they remain below the regular counts for a long time [17]. Moreover, a lack of advanced forms of spermatids has been demonstrated, as well as a significant increase in the rate of apoptosis of spermatogenic cells after the administration of nandrolone [18,19]. The fluorescent, in situ hybridization of sperm revealed disomies of XY and chromosomes 1 and 9, which suggests that the use of anabolic–androgenic steroids is associated with anomalies in the meiotic process and genetic damage [20]. The results of these studies indicate that, despite the return of spermatogenesis after the discontinuation of anabolic–androgenic steroids, fertility may remain permanently impaired. This is confirmed by the results of Al Hashimi’s research, which showed a high rate of abnormal semen findings (79%) and a considerable percentage of patients with infertility (18%). In patients with infertility, only 15% reported a successful pregnancy before the end of the 12-month follow-up period, despite treatment, indicating the long-term negative influence of anabolic–androgenic steroid abuse on fertility [21].

In addition to its direct impact on fertility, androgen abuse is also associated with many other adverse effects that may indirectly affect fertility. Common side effects include acne, mainly of the trunk, and gynecomastia [12]. The most serious side effect associated with any 17α-alkylated androgens (the main class of orally active synthetic androgens, as well as with SARMs, a new structurally derived antiandrogenic non-steroidal androgens) is hepatotoxicity [22,23,24]. Hepatotoxicity can have a variety of clinical manifestations, including liver tumors, adenoma, cancer, cholangiocarcinoma, angiosarcoma, peliosis hepatis, and cholestasis [25,26]. Peliosis hepatis, focal hepatic necrosis with vascular cyst formation, even if it is benign, leads to the enlargement of the liver and spleen, and can lead to fatal bleeding, both spontaneous and after liver biopsy [27]. Other serious side effects include cerebral- or deep-vein thrombosis, pulmonary embolism, cerebral hemorrhage, and seizures [28], as well as HIV and hepatitis from sharing needles, although this is less common for androgen abusers than for other injecting drug users [29,30]. The side effects of the injections can also include local injury and sepsis [12]. Overwork injury, rhabdomyolysis, and popliteal entrapment may also be associated with androgen abuse [31,32].

Psychological effects are among the distinct side effects of androgen abuse. Androgens dose-dependently influence the mood and induce hypomania in healthy subjects [33]. When high doses of or very strongly acting androgens are used, androgen abuse is associated with increased impulsivity, aggression, violence, or dysphoria, including depression or anergy and the acceleration of psychosis. Typical for androgen abusers are addictive behaviors such as tolerance, reinforcement, withdrawal symptoms, and craving-driven drug-seeking behaviors [33]. In addition, a few cardiovascular diseases like hypertension, cardiomyopathy, dyslipidemia, or prostatic effects of high-dose androgens have been annumerated [33,34].

Another threat stems from the fact that androgen abusers often purchase counterfeit or veterinary preparations frequently obtained through illegal sale via underground networks. Because the composition and purity of these preparations are unknown, their impact on the body is difficult to predict [12]. Only a small proportion of androgen comes from a health professional. According to some sources, approximately 21% of users report health professionals as their primary source of androgens [8]. In many cases, these are drugs from the pharmacy market purchased abroad, which may be of uncertain composition and purity.

Taking into account the fact that the percentage of infertile men in Central and Eastern Europe is among the highest in the world, and that the illegal use of testosterone significantly reduces male fertility, the purpose of the present study was to identify the prevalence of testosterone use in recreational athletes recruited from gyms randomly selected in Wrocław, as well as factors that are associated with the use of testosterone, taking into account the issue of having children and reproductive plans. Additionally, this study aimed to attempt to determine the factors related to the use of testosterone and anabolic–androgenic substances and to create a profile of a person using testosterone, which may enable the implementation of the obtained information in preventive activities regarding the use of testosterone and anabolic–androgenic steroids in a population with such characteristics.

## 2. Materials and Methods

### 2.1. Study Design

#### 2.1.1. Method Used

A cross-sectional analysis of data collected from an anonymous survey was performed, encompassing athletes practicing physique and strength sports, recruited from randomly selected gyms in Wrocław.

#### 2.1.2. Recruitment of Respondents

The respondents were recruited by posting a survey on the Facebook pages of public gyms between June 2022 and June 2023.

#### 2.1.3. Admission Criteria

The criteria for selecting the gym included the location of the gym in Wrocław and a possibility of conducting a survey using the gym’s Facebook platform. The inclusion criteria for this study were as follows: male gender, aged over 18 years, and the ability to understand each element of the survey. Since the survey was anonymous and conducted online, the fulfillment of the above criteria was checked based on the respondents’ answers. In the survey, the participants were asked if all of the questions were absolutely clear and if they had any comments on the survey. If any respondent had stated that any of the questions were unclear to him, his participation in this study would have been cancelled. However, such an event did not take place.

This study involved people regardless of skin color, class or social group, and both professional and non-professional athletes.

The way the respondents were included in the control and research groups was as follows: the control group consisted of the respondents who declared in the survey that they did not use testosterone, and the research group consisted of people who declared that they used testosterone.

### 2.2. The Questionnaire Structure

To assess the current scale of the illegal use of testosterone, as well as the current state of knowledge about its use and possible side effects, the behaviors associated with it, and the characteristics of its users, an original questionnaire was created.

The questionnaire consisted of 32 questions divided into three parts. The first part (13 questions addressed to each respondent) contained questions concerning the socio-demographic information and data on training habits and competition participation. The second part (14 questions, addressed only to the individuals using testosterone) contained the following: questions about the use of testosterone, its duration, the reasons for its use, the knowledge of testosterone and other androgens, the simultaneous use of other androgens and other substances and the reasons for their use, medical consultations related to the use of testosterone and the purpose of these consultations, blood tests while using testosterone, and knowledge of the side effects of testosterone use. The third part (5 questions, only for the respondents not taking testosterone) contained questions about whether the respondents had ever considered using testosterone, whether they had been encouraged to use it, whether they were considering its use in the future, and the reason for their choice not to use testosterone. 

The questions were mainly closed, and, in justified cases, open. This mainly concerned questions about the additional substances used together with testosterone and it made it possible to not include all possible trade names in the question, allowing the respondents to provide the trade names themselves. The open questions also allowed the participants to indicate other possible laboratory tests and purposes of medical consultations other than the most frequently performed ones listed in the questionnaire, as well as the additional motivational factors for the use of testosterone and other concomitantly used substances.

The yes/no questions and questions concerning socio-demographic data were of multiple choice. The remaining questions were multiple-choice as well. The survey was collected online. The interviewer was not present when the questionnaire was being completed so as not to influence the sincerity of the answers.

### 2.3. Statistical Analysis

Based on the “Sample size determination for logistic regression revisited,” published by Demidenko (2007), we calculated the minimal sample size for the univariate logistic regression model [35]. A two-sided test was used with the OR = 2, Pr(H0) = 0.4, and α for type I error probability was set at α = 0.05, while the β type II error probability was β = 0.2. The calculated sample size was 153 observations. The calculation was performed with the G*Power, 3.1 version.

The categorical variables were presented as percentages. To examine the relationship between the use of testosterone and the socio-demographic information (obtained to create a profile of the testosterone user), the univariate logistic regression model was performed. The odds ratio (OR) with a 95% confidence interval was calculated.

The statistical significance was defined as a two-sided probability value of less than 0.05. The calculation was performed with the TIBCO Software Inc., (Palo Alto, CA, USA) (2017) Statistica (data analysis software system), version 13.

## 3. Results

### 3.1. Socio-Demographic Information, Data on Training Habits, and Competition Participation

The socio-demographic information, data on training habits, and competition participation are presented in Table 1.

#### 3.1.1. Age

The majority of the respondents were aged 18–40. The largest groups included people aged 18–25 years (49.72%) and aged 26–30 years (26.26%). In the group of people using testosterone and in the group not using testosterone, these two age groups were also most represented (33.33% and 58.62%, respectively) (Table 1).

#### 3.1.2. Education

Most of the respondents declared higher or secondary education (49.16% and 45.25%, respectively). In the group of people using testosterone, the majority were people with secondary education (52.38%), and the second largest group consisted of people with higher education (41.27%). In the group of people not using testosterone, the majority consisted of people with higher education (53.45%), and the second largest group consisted of people with secondary education (41.38%) (Table 1).

#### 3.1.3. Economic Status

Among the total number respondents, as well as among the people using testosterone and not using testosterone, people with middle-class earnings dominated (46.37%, 50.79%, and 43.97%, respectively) (Table 1).

#### 3.1.4. Employment

The majority of the respondents were students (24.58%), the second largest group consisted of people running their businesses (17.88%), and the third largest group consisted of corporate employees (16.76%). Among the people using testosterone, people running their businesses predominated (25.4%), followed by corporate employees and people in managerial positions in corporations and enterprises (15.87% and 14.29%). Among the people who did not use testosterone, students dominated (33.62%). The next largest groups were corporate employees, people running their own businesses, and those employed by medium-sized and small-sized enterprises (17.24%, 13.79%, and 13.79%, respectively) (Table 1).

#### 3.1.5. Marital Status

Among all of the respondents and in the group of people using testosterone, people in permanent partnerships dominated (44.69% and 46.03%, respectively). The second position was occupied by single people (40.22% and 31.75%, respectively). In the group of people not using testosterone, the majority were single people (44.83%). The second position was taken by men in a permanent relationship (43.97%) (Table 1).

#### 3.1.6. Sexual Orientation

The majority of the respondents in general and in the study and control groups declared heterosexual orientation (96.9%, 96.83%, and 95.69%, respectively) (Table 1).

#### 3.1.7. Domicile

The majority of the respondents in general and in the study and control groups declared that they lived in a city with over 250,000 inhabitants (68.16%, 53.97%, and 75.86%, respectively) (Table 1). 

#### 3.1.8. Length of Training Experience

The majority of the respondents in general and in the study and control groups declared that they had been training for over 24 months (79.89%, 93.65%, and 72.41%, respectively) (Table 1).

#### 3.1.9. Frequency of Training

The majority of the respondents in general and in the study and control groups declared that they trained 3–5 times a week (55.31%, 68.25%, and 48.28%, respectively) (Table 1). 

#### 3.1.10. Participation in Weightlifting Competitions

The majority of the respondents in general and in the study and control groups declared that they did not compete in weightlifting competitions (77.65%, 66.67%, and 83.62%, respectively) (Table 1).

#### 3.1.11. Participation in Bodybuilding Competitions

The majority of the respondents in general and in the study and control group declared that they did not compete in weightlifting competitions (96.65%, 90.48%, and 100%, respectively) (Table 1). 

#### 3.1.12. Having Children

The majority of the respondents in general and in the study and control groups declared that they had children (83.8%, 76.19%, and 87.93%, respectively) (Table 1).

#### 3.1.13. Planning to Have Children

The majority of the respondents in general and in the study and control groups declared that they planned to have children (59.22%, 52.38%, and 62.93, respectively) (Table 1).

### 3.2. Data Regarding Testosterone Users

The data regarding testosterone users are presented in Table 2.

#### 3.2.1. Use of Testosterone or Other Anabolic–Androgenic Steroids

The use of testosterone or other anabolic–androgenic substances was declared by 35.2% of respondents. Of the people using testosterone or other anabolic–androgenic steroids, over half (50.79%) declared that they had been using them for more than 12 months. A total of 22.2% declared the duration of use for as 3–6 months. A total of 14.29% of the testosterone users declared the duration of use as 1–3 months and 11.11% as 6–12 months (Table 2).

#### 3.2.2. Purpose of Using Testosterone or Other Anabolic–Androgenic Steroids

The most common answers were as follows: improvement of training effects (87.3%), improvement of body shape (60.32%), and improvement of resistance to heavy training (60.32%). All of the respondents’ answers are summarized in Table 2.

#### 3.2.3. Use of Other Substances

A total of 76.19% of testosterone users declared the simultaneous use of other substances. The substances used most often simultaneously with testosterone were selective estrogen receptor modulators (used by 75% of people using testosterone). The second place was occupied by aromatase inhibitors (declared by 56.2% of testosterone users). The third and fourth positions were occupied by gonadotropins and stimulants (47.92% each) (Table 2).

#### 3.2.4. Purpose of Using Other Substances

The most common reasons for using other substances at the same time included increasing the effect of testosterone (95.83%), reducing testosterone side effects (68.75%), and improving the body shape (31.25%). All of the answers provided by the respondents regarding the purpose of using substances other than testosterone are summarized in Table 2.

#### 3.2.5. What Prompted the Respondents to Use Testosterone or Other Anabolic–Androgenic Steroids?

The most common factors that prompted the respondents to use anabolic–androgenic steroids included persuasion from friends (43.33%), the Internet (35%), publications (31.67%), and a personal coach (18.33%). The list of all of the factors is summarized in Table 2.

#### 3.2.6. Sources of Knowledge about Testosterone and Other Anabolic–Androgenic Steroids

The most frequently declared sources of knowledge about testosterone and other anabolic–androgenic steroids included the Internet (76.19%), scientific publications (66.67%), friends (38.1%), and a personal coach (31.75%). A summary of all of the declared sources of knowledge about testosterone and other anabolic–androgenic steroids is summarized in Table 2.

#### 3.2.7. Medical Consultations Regarding the Use of Testosterone and Other Anabolic–Androgenic Steroids

The majority (60.32%) of people using testosterone and other anabolic–androgenic steroids did not consult a doctor in connection with their use (Table 2).

#### 3.2.8. Purpose of Medical Consultations Regarding the Use of Testosterone and Other Anabolic–Androgenic Steroids

The most common reasons why the respondents using testosterone and other anabolic–androgenic steroids consulted a doctor in connection with the use of these substances included the following: the willingness to receive a referral for laboratory tests (66.67%), a discussion of the side effects of their use (44.44%), advice on how to counteract the side effects that occurred (44.44%), and trying to obtain a prescription for testosterone or other preparations used (25.93%). The entire list is presented in Table 2.

#### 3.2.9. Performing Laboratory Tests in Connection with the Use of Testosterone and Other Anabolic–Androgenic Steroids

The majority of respondents using testosterone or other anabolic–androgenic substances (92.06%) performed laboratory tests in connection with their use (Table 2).

#### 3.2.10. Type of Laboratory Test Performed in Connection with the Use of Testosterone and Other Anabolic–Androgenic Steroids

The laboratory tests most frequently performed during the use of testosterone and other anabolic–androgenic steroids were morphology (91.53%), estradiol level (86.44%), and thyroid-stimulating hormone (TSH) level (84.75%). A list of the tests most frequently performed in connection with the use of testosterone and other anabolic–androgenic substances is presented in Table 2.

#### 3.2.11. Knowledge about the Complications of Using Testosterone and Other Anabolic–Androgenic Steroids

The most common complications of testosterone use known to the respondents using testosterone and other anabolic–androgenic steroids included the following: testicular shrinkage (87.10%), infertility (83.87%), hypertension (83.87%), breast enlargement (74.19%), and changes in blood count (70.97%). A summary of responses is presented in Table 2.

### 3.3. Data Regarding People Not Using Testosterone

The data regarding people not using testosterone are presented in Table 3.

#### 3.3.1. Planning to Use Anabolic–Androgenic Steroids

A total of 51.3% of people not using testosterone or other anabolic–androgenic steroids declared that they were planning to use testosterone in the future (Table 3).

#### 3.3.2. Considering the Use of Anabolic–Androgenic Steroids in the Future

A total of 31.3% of people not using testosterone or other anabolic–androgenic steroids declared that they were considering using testosterone in the future, and 22.61% might consider it (Table 3).

#### 3.3.3. Being Encouraged to Use Anabolic–Androgenic Steroids

The majority of the respondents (62.07%) were not encouraged to use testosterone or other anabolic–androgenic steroids (Table 3).

#### 3.3.4. Persons Encouraging the Use of Anabolic–Androgenic Steroids

The majority of the respondents encouraged to use testosterone were persuaded by their friends (77.27%) (Table 3).

#### 3.3.5. Factors That Influenced Decisions to Not Use Anabolic–Androgenic Steroids

The most common reasons for not using testosterone were the lack of such need (49.12%) and the fear of side effects (40.35%) (Table 3).

### 3.4. The Association between Socio-Demographic Data and Using Testosterone

The association between socio-demographic data and using testosterone is presented in Table 4.

Significant associations between testosterone and other anabolic–androgenic steroid intake and the other variables were shown for age, earnings, occupation, marital status, having children, length of training experience, frequency of training, and participation in weightlifting competitions.

Regarding age, men aged 26–30 were 2.62 times more likely to use testosterone and other anabolic–androgenic steroids than men aged 18–25, and men aged 31–35 were 3.24 times more likely to use them. As for earnings, persons with earnings at the upper-class level were 5.5 times more likely to use testosterone and other anabolic–androgenic steroids than men with no income, and persons with earnings at the middle-class level were 3.61 times more likely to use them (Table 4). As far as occupation is concerned, the most likely to use testosterone and other anabolic–androgenic steroids were managers (17.55 times more than students) (Table 4). Considering marital status, married persons were 2.8 times more likely to use testosterone and other anabolic–androgenic steroids than single men (Table 4). When taking into account having children, men with children were 2.28 times more likely to use testosterone and other anabolic–androgenic steroids than those without children (Table 4). As to the length of training and frequency of training, the most likely to use testosterone and other anabolic–androgenic steroids were men training for over 24 months (2.81 × 10^6^ times more than those training for less than 3 months) and training 5–7 times a week (3.38 × 10^7^ times more than those training for less than a few times a month) (Table 4). Moreover, men participating in weightlifting competitions were 2.55 times more likely to use testosterone and other anabolic–androgenic steroids than those not participating in competitions (Table 4).

## 4. Discussion

The results of our study, based on an anonymous survey of 179 people recruited from gyms randomly selected in Wrocław, showed that more than 35% of the respondents used testosterone at the time of the study. To our knowledge, no other studies evaluating the illegal use of testosterone among bodybuilders and strength athletes in Poland exist, and we cannot compare our results with other Polish data. If compared with the results obtained by Montuori et al., a study conducted as a cross-sectional analysis of data from an anonymous survey of 107 bodybuilders recruited from 30 gyms randomly selected in the metropolis of Naples [11], we can conclude that the frequency of testosterone use in Wrocław in Poland was similar (35.2% in our study vs. 35.51% found by Montuori et al.). Although the respondents of the Montuori et al. study declared the use of hormones in general, not specifically testosterone or other anabolic–androgenic steroids, anabolic–androgenic steroids are the most frequently used hormones among strength athletes. They are usually used in the form of cycles composed of several substances, with testosterone as the basic component, and we can speculate that the majority of people declaring the use of hormones in the study by Montuori et al. used testosterone and other anabolic–androgenic steroids. Since both studies were performed in recent years and the data were collected in large European cities, one can assume that they reflect the current situation regarding the use of these substances in large European centers.

The meta-analysis of 271 articles on the prevalence of androgen abuse by Sagoe et al. revealed that lifetime androgen use in recreational sports constituted 18.4% of all of the subjects, which is a much lower result than that of our study [7]. Similar results were obtained by Alsaeed et al. In their cross-sectional survey of ten fitness centers in Kuwait, the authors showed that 22.7% of 194 respondents used anabolic–androgenic steroids [36]. Nevertheless, these studies were published in 2014 and 2015, and the differences between them and our study may be due to the growing trend towards the illegal use of anabolic–androgenic steroids. In addition, the studies concerned all people who practice sports more recreationally, not specifically body and strength sports. Our results also differ from those of a similar study published in 2017, conducted with a group of 457 people who train regularly in gyms in Riyadh, Saudi Arabia, where 7.9% of the subjects reported taking hormones [37]. Similarly, in another study conducted in Teheran, Iran, approximately 16.6% of recreational athletes had a history of anabolic–androgenic steroid abuse [38]. A study of gym users in Al Ain, United Arab Emirates, showed a prevalence of the misuse of anabolic steroids reaching 22% [39]. Perhaps the discrepancies in the percentage of people using testosterone illegally result from cultural differences and different approaches to muscular appearance.

The growing trend in the illegal use of testosterone is also indirectly confirmed by our results, which showed that 51.3% of individuals not using testosterone planned to use it in the future, and 31.3% considered it. This trend was not so clear in 2015. Then, only 10.3% of the respondents surveyed by Alsaeed et al. declared their willingness to use anabolic–androgenic steroids in the future, and 43.4% did not plan to use them [36]. The increasing number of men using testosterone indicates unfavorable lifestyle changes that may significantly affect male fertility. This is particularly disturbing considering the profile of the person using testosterone, which we outlined based on the survey that we had conducted. Namely, a typical person using testosterone in the population of people training amateurly in gyms in Wrocław, Poland, is a man aged 26–35, whose earnings are at the level of middle class or higher, who is married, has children, has training experience of at least six months, exercises at least once a week, takes part in weightlifting competitions, is a manager in a corporation or enterprise, runs his own business or is an employee of a corporation, or is employed in a small- or medium-sized enterprise. There was no significant association between testosterone use and education, sexual orientation, or the size of the town where the respondents lived. Regarding the age of the people using testosterone and other anabolic–androgenic steroids, the results additionally showed that men aged 26–30 are 2.62 times more likely to use them than men aged 18–25, and men aged 31–35 are 3.24 times more likely to use them than men aged 18–25. Our results are in line with the results of Al Hashimi et al. [40]. The authors, with an online survey composed of 30 questions conducted among urologists, andrologists, and endocrinologists, showed that the most common age categories of anabolic–androgenic steroids users were 20–30 years (74.83%) and 30–40 years (13.25%) [40]. Similarly, Montuori et al. showed that the average age of hormone users was 32.42 years [11].

Additionally, our study showed that there is a relationship between earnings and the likelihood of respondents using testosterone and other anabolic–androgenic steroids. Upper-class income earners were 5.50 times more likely to use them than those with no income, and middle-class income earners were 3.61 times more likely to do so than those with no income. Moreover, managers were 17.5 times more likely to use testosterone and other anabolic–androgenic steroids than students, and married men were 2.6 times more likely to use them than single men, while people with children were 0.65 times more likely to use these illegal substances than single men.

These characteristics regarding age, earnings, employment, and marital status correspond to people who are at the optimal age and have optimal socio-demographic conditions for having children, and the fact that people with such characteristics are most likely to use testosterone and other anabolic–androgenic steroids indicates that it is a significant social problem in the context of reproduction.

It is also worth noting that the men with a long training experience, training for 24 months, were 2.81 × 10^6^ times more likely to use testosterone and other anabolic–androgenic steroids than those training for less than 3 months. Similarly, men training systematically, 5–7 times weakly, were 3.38 × 10^7^ times more likely to use them than those who trained less than a few times a month. Our findings are consistent with the results of Montuori et al., which showed that most people used hormones to improve their body shape, most trained frequently, and approximately 50% reported practicing bodybuilding for longer than five years [11]. They are also consistent with the findings of Alsaeed et al., which showed that 81.8% of anabolic–androgenic steroid users and 71.4% of non-users who wanted to use it declared that achieving a muscular body shape was their priority [36]. In light of our results and those obtained in other studies, it can be concluded that the promotion of a perfect body shape by the media, on the one hand, contributes to regular training, but, on the other hand, may also generate behaviors that are harmful to health to achieve the intended goal, including the use of anabolic–androgenic steroids. This may, in turn, indirectly reduce fertility.

What is also worrying is the fact that the most common direct reason for using testosterone and other anabolic–androgenic steroids was the persuasion by friends. In addition, people who did not use illegal supplements were encouraged to use them mainly by their friends. This may indicate that people using testosterone consider its use effective but also relatively safe, since they actively engage in the encouragement process. This may be due to the wide availability of information on the use of testosterone and other anabolic–androgenic substances, their side effects, and the methods of preventing or alleviating them. Namely, over 3/4 of the respondents stated that they obtained knowledge about illegal assistance from the Internet, and over 2/3 declared that the source of knowledge was publications. The side effects known to the respondents, the selection of additional substances used simultaneously with the use of testosterone and other anabolic–androgenic steroids, and the type of laboratory tests performed indicate pretty extensive knowledge in this area.

Over 76% of testosterone users used other pharmacological agents. Most of them, indeed, declared that they did this to increase its anabolic effect; moreover, a reduction in testosterone side effects was in 2nd position, but the frequency distribution of the simultaneous use of other substances, of which selective estrogen receptor modulators accounted for 75%, aromatase inhibitors for 56.25%, and gonadotropins for 47.92%, indicates that they were used mainly to eliminate or alleviate the side effects of illegal support. This corresponds to the most frequently mentioned side effects known by the respondents, such as the following: a reduction in the size of testicles, infertility, hypertension, and breast enlargement, because these drugs are supposed to counteract these side effects (apart from hypertension). Alsaeed et al., in their study from 2015, showed that anabolic–androgenic steroid usage was associated with a similar (80%) frequency of concurrent use of other illicit substances [36]. The difference concerned the type of substances because, in the study by Alsaeed et al., ergogenic substances dominated.

Most testosterone users (over 92%) performed laboratory tests while using the illegal supplements. Surprisingly, most of them pointed to specific laboratory tests that they had performed in connection with the use of testosterone and other anabolic–androgenic steroids, despite their lack of medical education. The most common tests included blood count, estradiol, TSH, total and free testosterone, and liver function. The fact that they were able to indicate specific types of laboratory tests and such a distribution of answers may also indicate that they have relatively broad knowledge about the use of testosterone and other anabolic–androgenic steroids; however, it cannot be ruled out that that such a distribution of answers was random. In the study by Alsaeed et al., only 18.2% of the study respondents had appropriate knowledge of the side effects of anabolic–androgenic steroids [36]. The results of our study may indicate an increased awareness of the harmful effects of illegal doping, which may stem from the increased availability of information on steroid cycle patterns and the treatment of side effects.

Over 60% of testosterone users did not undergo medical consultations in connection with its use, and over 2/3 of the consultations were aimed at obtaining a referral for laboratory tests. This may indicate that the increase in access to information may be related to the respondents’ belief in their having sufficient knowledge about the safe use of illegal aids and the ability to cope with adverse symptoms on their own; however, other reasons cannot be ruled out, such as the fear of revealing the use of illegal aids. Regardless of the motivation, undertaking treatment and counteracting the side effects of testosterone and other anabolic–androgenic steroids on one’s own is not possible. The use of substances from the group of selective estrogen receptor modulators, aromatase inhibitors, and gonadotropins is associated with the possibility of the occurrence of side effects that are harmful to the gonads and the worsening of fertility disorders.

Our work has its limitations. Firstly, the research was conducted in one large city and is, therefore, not representative of the entire country. Secondly, based on surveys, we assess the current scale of the phenomenon of illegal use of testosterone and other anabolic–androgenic steroids and the behaviors associated with it, as well as the characteristics of the people using them. The survey form we have used allowed only for the estimation of behaviors that may potentially reduce fertility, but not the reduction in fertility itself. Although the profile of a man using testosterone that we established corresponded to people with the best socio-demographic conditions for having children, most of our respondents were young, and, currently, more and more people are postponing the decision to have children until later in life. Whether the disturbing trend we have demonstrated will translate into fertility reduction requires further investigation.

## 5. Conclusions

To sum up, the results of our study indicate a growing trend for the use of testosterone and anabolic–androgenic steroids and an increasing trend for gaining knowledge about these substances and their possible side effects, which unfortunately goes hand in hand with an unfavorable trend towards self-diagnosis and self-treatment of these effects. Additionally, this indicates that the profile of the person most likely to use testosterone corresponds to the characteristics of men in optimal socio-economic conditions for procreation. These results indicate that this practice is a significant social problem that may have a negative impact on male fertility. At the same time, creating a profile of the person most susceptible to the use of testosterone and other anabolic–androgenic steroids allows for preventive activities, which should include raising awareness of the irreversibility of some of the effects of testosterone and other anabolic–androgenic steroids and drugs such as selective estrogen receptor modulators, aromatase inhibitors, and gonadotropins on the gonads and possible permanent consequences of their use in the form of permanent fertility impairment and the permanent dysfunction of other organs. Additionally, it is important to build awareness that the purpose of training is to derive primarily health, physical, and mental benefits, and not only aesthetic ones. This approach usually excludes shortcuts using illegal means. According to the results of our study, information activities should be conducted using the tools that the respondents indicated as the main sources of knowledge about testosterone and other anabolic–androgenic steroids, i.e., via the Internet and publications addressed to this group of recipients. At the same time, the awareness of the importance of this problem should be increased among doctors, communities, and organizations that have tools for preventive activities, such as anti-doping agencies and organizations dealing with health promotion and public health.

## Figures and Tables

**Table 1 jcm-13-03719-t001:** Data on socio-demographic issues, training habits, and competition participation.

Variable	Levels	All Population (N = 179)	Study Group (N = 63)	Control Group (N = 116)
Age	18–25 years old	49.72	33.33	58.62
	26–30 years old	26.26	33.33	22.41
	31–35 years old	8.94	12.70	6.90
	36–40 years old	7.82	11.11	6.03
	41–45 years old	3.91	4.76	3.45
	46–50 years old	2.79	4.76	1.72
	51–55 years old	0.00	0.00	0.00
	56–60 years old	0.56	0.00	0.86
Education	Higher education (PhD or higher)	1.68	1.59	1.72
	Higher education	49.16	41.27	53.45
	Secondary education	45.25	52.38	41.38
	Vocational education	1.68	3.17	0.86
	Primary education	2.23	1.59	2.59
Earnings	Earnings above PLN 9404 gross/month	25.14	34.92	19.83
	Earnings PLN 3526.50 gross to PLN 9404 gross/month	46.37	50.79	43.97
	Earnings below PLN 3526.50 gross	13.41	7.94	16.38
	No income	15.08	6.35	19.83
Occupation	Students	24.58	7.94	33.62
	People running their own business	17.88	25.40	13.79
	Corporate employees	16.76	15.87	17.24
	Employees of small- and medium-sized enterprises	13.41	12.70	13.79
	Managers in corporations and enterprises	7.26	14.29	3.45
	Health care workers	6.70	3.17	8.62
	Civil servants	4.47	4.76	4.31
	Unemployed	2.79	3.17	2.59
	Other	2.23	6.35	0.00
	Coaches	1.12	1.59	0.86
	Engineers	1.12	3.17	0.00
	Uniformed services	1.12	1.59	0.86
	Athletes	0.56	0.00	0.86
Marital status	Married	15.08	22.22	11.21
	In a permanent partnership	44.69	46.03	43.97
	Single—not in a permanent relationship	40.22	31.75	44.83
Sexual orientation	Heterosexual	96.09	96.83	95.69
	Homosexual	2.79	3.17	2.59
	Bisexual	0.56	0.00	0.86
	Other	0.56	0.00	0.86
Domicile	Village	8.38	9.52	7.76
	City up to 50 thousand inhabitants	10.61	15.87	7.76
	City up to 100 thousand inhabitants	8.94	14.29	6.03
	City up to 250 thousand inhabitants	3.91	6.35	2.59
	City over 250 thousand inhabitants	68.16	53.97	75.86
Length of training experience	>24 months	79.89	93.65	72.41
	12–24 months	6.15	3.17	7.76
	6–12 months	7.26	3.17	6.48
	3–6 months	3.35	0.00	5.17
	<3 months	3.35	0.00	5.17
Frequency of training	5–7 times a week	8.38	12.70	6.03
	3–5 times a week	55.31	68.25	48.28
	2–3 times a week	30.17	19.05	36.21
	Once a week	2.79	0.00	4.31
	A few times a month	2.79	0.00	4.31
	Less than a few times a month	0.56	0.00	0.86
Participation in weightlifting competitions	No	77.65	66.67	83.62
	Yes	22.35	33.33	16.38
Participation in bodybuilding competitions	No	96.65	90.48	100.00
	Yes	3.35	9.52	0.00
Having children	No	83.80	76.19	87.93
	Yes	16.20	23.81	12.07
Planning to have children	No	40.78	47.62	37.07
	Yes	59.22	52.38	62.93

Test group—people using testosterone; Control group—people not using testosterone; N—number of respondents. Values are expressed in % of N.

**Table 2 jcm-13-03719-t002:** Data regarding testosterone users.

Variable	Levels	Frequency % (k)
Testosterone use	No	64.80 (116)
(N = 179)	Yes	35.20 (63)
Testosterone use period (*n* = 63)	<1 month	1.59 (1)
	1–3 months	14.29 (9)
	3–6 months	22.22 (14)
	6–12 months	11.11 (7)
	>12 months	50.79 (32)
Purpose of using testosterone and other anabolic–androgenic steroids (*n* = 63)	Improving training effects	87.3 (55)
	Improving resistance to heavy training	60.32 (38)
	Improving body shape	60.32 (38)
	Improving well-being	44.44 (28)
	Improving libido	25.4 (16)
	Supplementation of testosterone deficiency	3.17 (2)
	Improving sleep and regeneration	1.59 (1)
	Body weight reduction	1.59 (1)
	All of the above and improved health	1.59 (1)
	Improving results in competitions	1.59 (1)
Concomitant use of other anabolic–androgenic steroids (*n* = 62)	No	22.58 (14)
	Yes	77.42 (48)
Substances used simultaneously with testosterone (*n* = 48)	Selective estrogen receptor modulators	75 (36)
	Aromatase inhibitors	56.25 (27)
	Gonadotropins	47.92 (23)
	Stimulants	47.92 (23)
	Hormones	35.42 (17)
	Anabolic–androgenic steroids	31.25 (15)
	Other	12.5 (6)
	Dietary supplements	6.25 (3)
	Dopamine agonist	6.25 (3)
	Selective androgen receptor modulators	6.25 (3)
Reason for simultaneous use of other substances (*n* = 48)	Enhancing the effects of testosterone	95.83 (46)
	Reducing the side effects of testosterone	68.75 (33)
	Increasing only lean body mass	31.25 (15)
	Reducing estradiol levels	4.17 (2)
	Reducing prolactin levels	2.08 (1)
	Improving well-being	2.08 (1)
	Counteraction of testicular atrophy	2.08 (1)
	Normalization of hormonal test results	2.08 (1)
	Improving body shape	2.08 (1)
	Increasing strength and efficiency	2.08 (1)
What prompted respondents to use testosterone and other anabolic–androgenic steroids (*n* = 60)	Friends	43.33 (26)
	Internet	35 (21)
	Publications	31.67 (19)
	Personal coach	18.33 (11)
	Own decision	5 (3)
	Lack of improvement in training effects without anabolic–androgenic steroids	3.33 (2)
	A wish to improve recovery after exercise	
	Low testosterone level	3.33 (2)
	Family doctor	3.33 (2)
	Sports doctor	1.67 (1)
	A wish to move to a higher training level	1.67 (1)
	Bad mood	1.67 (1)
	A wish to try testosterone and other	1.67 (1)
	anabolic–androgenic steroids	1.67 (1)
	out	
	A wish to do sports professionally	1.67 (1)
	Knowledge	1.67 (1)
	Ambition	1.67 (1)
Source of knowledge about testosterone (*n* = 63)	Internet	76.19 (48)
	Publications	66.67 (42)
	Friends	38.1 (24)
	Personal coach	31.75 (20)
	Doctor	3.17 (2)
	Own experience	1.59 (1)
	Scientific investigation	1.59 (1)
	Scientific information on the Internet	1.59 (1)
	Podcasts	1.59 (1)
Medical consultation regarding the use of testosterone and anabolic–androgenic steroids (*n* = 62)	No	61.29 (38)
	Yes	38.71 (24)
Purpose of medical consultation (*n* = 27)	To obtain a referral for check-up tests	66.67 (18)
	Discussion of side effects	44.44 (12)
	Discussion of side effects associated with the use of testosterone/other anabolic–androgenic steroids	44.44 (12)
	To obtain a prescription for testosterone or other medications	25.93 (7)
	To obtain medical advice regarding the duration of use of testosterone/other anabolic–androgenic steroids	18.52 (5)
	Assessment of the risk of low testosterone levels	18.52 (5)
	Dosage advice	18.52 (5)
Performing laboratory tests in connection with the use of testosterone and other anabolic–androgenic steroids (*n* = 61)	No	4.92 (3)
	Yes	95.08 (58)
Type of laboratory test (*n* = 59)	Blood count	91.53 (54)
	Estradiol	86.44 (51)
	TSH	84.75 (50)
	Liver tests	81.36 (48)
	Total testosterone	81.36 (48)
	Free testosterone	79.66 (47)
	Lipid profile	72.88 (43)
	Thyroid	71.19 (42)
	Free triiodothyronine	71.19 (42)
	Free thyroxine	71.19 (42)
	SHBG	52.54 (31)
	Fasting glucose	44.07 (26)
	Fasting insulin	25.42 (15)
	DHEA-S	23.73 (14)
Awareness of possible side effects of the use of testosterone and other anabolic–androgenic steroids (*n* = 62)	Reduction in the size of testicles	87.1 (54)
	Infertility	83.87 (52)
	Hypertension	83.87 (52)
	Breast enlargement	74.19 (46)
	Changes in blood count	70.97 (44)
	Prostatic hyperplasia	69.35 (43)
	Liver damage	66.13 (41)
	Thrombosis	50 (31)
		46.77 (29)

N—number of respondents, *n*—number of people who answered a given question, k—number of people who provided a given answer. Values are expressed in % of N or n.

**Table 3 jcm-13-03719-t003:** Data regarding people not using testosterone.

Variable	Levels	Frequency % (k)
Planning to use of anabolic–androgenic steroids (N = 115)	No	48.70 (56)
	Yes	51.30 (59)
Being encouraged to use anabolic–androgenic steroids (N = 116)	No	62.07 (72)
	Yes	37.93 (44)
Persons encouraging the use of anabolic–androgenic steroids (N = 44)	Friends from the gym/sports club	77.27 (34)
	Social groups–Internet	18.18 (8)
	Personal coach	2.27 (1)
	Partner	2.27 (1)
Factors that influenced decisions not to use anabolic–androgenic steroids (N = 114)	Lack of need	49.12 (46)
	Fear of side effects	40.35 (29)
	Confidence in one’s own capabilities	5.26 (27)
	Too young age	1.39 (1)
	Beliefs	0.88 (1)
	Partner	0.88 (1)
	Lack of availability	0.88 (1)
	Fear of detection of the use of these substances during sports competitions	0.88 (1)
Considering the use of anabolic–androgenic steroids in the future (N = 115)	No	46.09 (53)
	Maybe	22.61 (26)
	Yes	31.30 (36)

N—number of people who answered a given question, k—number of people who provided a given answer. Values are expressed in % of N.

**Table 4 jcm-13-03719-t004:** The association between socio-demographic data and testosterone use.

Variable	Levels	OR	95% CL	*p*-Value
**Age**	18–25 years old (ref.)	-	-	-
	**26–30 years old**	**2.62**	**1.23–5.56**	**0.013**
	**31–35 years old**	**3.24**	**1.08–9.68**	**0.036**
	**36–40 years old**	**3.24**	**1.02–10.29**	**0.046**
	41–45 years old	2.43	0.5–11.73	0.269
	46–50 years old	4.86	0.76–31.04	0.095
	56–60 years old	0.00	0–0	0.998
Education	Higher education (PhD or higher)	1.50	0.06–40.64	0.810
	Higher education	1.26	0.12–12.66	0.845
	Secondary education	02.06	0.21–20.7	0.538
	Vocational education	6.00	0.22–162.54	0.287
	Primary education (ref.)	-	-	-
**Earnings**	**Earnings above PLN 9404 gross/month**	**5.50**	**1.64–18.48**	**0.006**
	**Earnings PLN 3526.50 gross to PLN 9404 gross/month.**	**3.61**	**1.14–11.4**	**0.029**
	Earnings below PLN 3526.50 gross	1.51	0.36–6.44	0.575
	No income (ref.)	-	-	-
**Marital status**	**Married**	**2.80**	**1.12–6.99**	**0.027**
	In a permanent partnership	1.48	0.74–2.94	0.266
	Single–not in a permanent relationship (ref.)	-	-	-
Sexual orientation	Heterosexual	0.82	0.13–5.07	0.835
	Homosexual (ref.)	-	-	-
	Bisexual	0.00	0–0	0.998
	Other	0.00	0–0	0.998
Domicile	Village (ref.)	-	-	-
	City up to 50 thousand inhabitants	1.67	0.42–6.56	0.465
	City up to 100 thousand inhabitants	1.93	0.46–8.05	0.368
	City up to 250 thousand inhabitants	2.00	0.32–12.33	0.455
	City with over 250 thousand inhabitants	0.58	0.19–1.75	0.334
**Length of training** **experience**	**>24 months**	**2.81 × 10^6^**	**0–0**	**0.000**
	**12–24 months**	**8.90 × 10^5^**	**1.86 × 10^5^–4.27 × 10^6^**	**0.000**
	**6–12 months**	**7.28 × 10^5^**	**1.56 × 10^5^–3.41 × 10^6^**	**0.000**
	3–6 months	0.02	0–0	0.999
	<3 months (ref.)	-	-	-
**Frequency of training**	**5–7 Times A Week**	**3.38** **× 10^7^**	**1.02** **× 10^7^–1.12 × 10^8^**	**0.000**
	**3–5 Times A Week**	**2.27** **× 10^7^**	**1.07** **× 10^7^–4.83 × 10^7^**	**0.000**
	**2–3 Times A Week**	**8.45** **× 10^6^**	**0–0**	**0.000**
	Once A Week	0.14	0–0	1.000
	A Few Times A Month	0.14	0–0	1.000
	Less Than A Few Times A Month (ref.)	-	-	-
**Participation in** **weightlifting** **competitions**	No (ref.)	-	-	-
	**Yes**	**2.55**	**1.24–5.24**	**0.011**
Participation in bodybuilding competitions	No (ref.)	-	-	-
	Yes	4.45 × 10^8^	0–0	0.997
**Having children**	No (ref.)	-	-	-
	**Yes**	**2.28**	**1.02–5.09**	**0.045**
Planning to have children	No (ref.)	-	-	-
	Yes	0.65	0.35–1.21	0.171
**Occupation**	Other	1.71 × 10^9^	0–0	0.998
	Engineers	1.71 × 10^9^	0–0	0.998
	**Managers in corporations and enterprises**	17.55	**3.91–78.76**	**0.000**
	Coaches	7.80	0.42–145.21	0.169
	Uniformed services	7.80	0.42–145.21	0.169
	**People running their own business**	7.80	**2.44–24.9**	**0.001**
	Unemployed	5.20	0.69–39.08	0.109
	Civil servants	4.68	0.85–25.81	0.076
	**Corporate employees**	3.90	**1.17–12.96**	**0.026**
	**Employees of small- and medium-sized enterprises**	3.90	**1.11–13.75**	**0.034**
	Health care workers	1.56	0.26–9.26	0.625
	Athletes	0.00	0–0	0.999
	Students (ref.)	-	-	-

OR—odds ratio. Statistically significant *p*-values (<0.05) are in bold. Estimates were derived from univariate logistic regression models.

## Data Availability

The data supporting the findings of this study are available from the corresponding author on request.

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
