# Peer review of "Illegal Use of Testosterone and Other Anabolic–Androgenic Steroids in the Population of Amateur Athletes in Wrocław, Poland—An Unfavorable Lifestyle Trend in the Population of Men of Reproductive Age"

_jcm, 2024, doi:10.3390/jcm13133719_

Round 1
Reviewer 1 Report
Comments and Suggestions for Authors
The authors should be congratulated for their efforts. The topic is novel and intriguing. The recreational use is worrisome and as a physician being aware of this unsafe practice is mandatory. However, my biggest concerns are relative to how the data were collected. First, how did the authors verify this statement "being able to understand every item of the questionnaire"? They cannot say that. How did the recruitment work? Snowball recruitment (PMID= 38010890)? Web-based online survey? What were the proportions of each recruitment if there were more than one recruitment option? Moreover, how the control group was enrolled?
The results indicated that 35 % of respondents used testosterone. Which formulation? Were the side effects experimented in which percentage? What were the comorbidity of those patients? And the medications?
Moreover, the authors also identified that two main causes of testosterone self-administration were the "improvement of training effects" and the "improvement of body shape". What is the health policy that the authors would adopt to limit this uncontrolled phenomenon? 35% is a worrisome number that could provide an active strategy to reduce this phenomenon. The authors should properly address and even stress this main aspect in the discussion section.
Author Response
Reviewer #1
Comments and Suggestions for Authors
The authors should be congratulated for their efforts. The topic is novel and intriguing. The recreational use is worrisome and as a physician being aware of this unsafe practice is mandatory.
Response: We thank Reviewer #1 for their positive review. We are very glad to see the Reviewer found the paper topic novel and intriguing, and the results important for physicians.
We would like to thank the Reviewer #1 for taking time and effort to review the manuscript. We are very grateful for all valuable comments and suggestions that helped us further revise and improve the quality of the manuscript. Below, we respond point by point to the Reviewer's comments. The relevant corrections are highlighted in the revised manuscript bold and with track changes made of Word.
However, my biggest concerns are relative to how the data were collected. First, how did the authors verify this statement "being able to understand every item of the questionnaire"? They cannot say that. How did the recruitment work? Snowball recruitment (PMID= 38010890)? Web-based online survey? What were the proportions of each recruitment if there were more than one recruitment option? Moreover, how the control group was enrolled?
Response: We thank Reviewer #1 for these comments. Indeed, the description of how the results were obtained was incomplete and we greatly appreciate Reviewer #1 for pointing it out. As per the Reviewer's suggestion, we have developed this issue as below (page 4, lines 229-251).
„Method used.
A cross-sectional analysis of data collected from an anonymous survey was performed, encompassing athletes practicing physique and strength sports, recruited from gyms, randomly selected in Wrocław.
Recruitment of respondents.
Respondents were recruited by posting a survey on the Facebook pages of public gyms between June 2022 and June 2023.
Admission criteria.
The criteria for selecting the gym included: location of the gym in WrocÅ‚aw, a possibility of conducting a survey using the gym’s Facebook platform. Inclusion criteria for the study were as follows: male gender, age over 18, ability to understand each element of the survey. Since the survey was anonymous and conducted online, the fulfillment of the above criteria was checked based on the respondents' answers. In the survey the participants were asked if all the questions were absolutely clear and if they had any comments on the survey. If any respondent had stated that any of the questions were unclear to him, his participation in the study would have been cancelled. However, such an event did not take place.
The study involved people regardless of skin color, class or social group, both professional and non-professional athletes.
The way respondents were included in the control and research groups.
The control group consisted of respondents who declared in the survey that they did not use testosterone, and the research group consisted of people who declared that they used testosterone.”
The results indicated that 35 % of respondents used testosterone. Which formulation? Were the side effects experimented in which percentage? What were the comorbidity of those patients? And the medications?
Response: We thank Reviewer #1 for these questions. Due to differences only in the pharmacokinetics and not in the effects of different testosterone preparations, in our study we did not ask about the type of testosterone preparations used. The survey was designed to assess the scale of illegal use of testosterone and to create a profile of people using it, not its impact on the health of people using testosterone, therefore we did not ask questions about side effects and comorbidities. In the context of the scale of illegal use of testosterone and other anabolic-androgenic steroids, this is a very interesting topic for further research that will complement the results obtained in the current study.
Moreover, the authors also identified that two main causes of testosterone self-administration were the "improvement of training effects" and the "improvement of body shape". What is the health policy that the authors would adopt to limit this uncontrolled phenomenon? 35% is a worrisome number that could provide an active strategy to reduce this phenomenon. The authors should properly address and even stress this main aspect in the discussion section.
Response: We thank Reviewer #1 for this suggestion. Accordingly, the conclusions section was expanded as follows (pages 20-21, lines 822-858):
„Additionally, it is important to build awareness that the purpose of training is to derive primarily health, physical and mental benefits, and not only aesthetic ones. This approach usually excludes shortcuts using illegal means. According to the results of our study, information activities should be conducted using the tools that respondents indicated as the main sources of knowledge about testosterone and other anabolic-androgenic steroids, i.e. via the Internet and publications addressed to this group of recipients. At the same time, awareness of the importance of this problem should be increased among doctors, communities and organizations that have tools for preventive activities, such as anti-doping agencies and organizations dealing with health promotion and public health.”

Reviewer 2 Report
Comments and Suggestions for Authors
Dear authors, after reading your manuscript, I believe it has many drawbacks:
1. Firstly, the title is not appropriate, you did not actually prove that illegal use of testosterone affected your subjects fertility status. It is true that testosterone use is linked to male infertility, you have not evaluated the fertility status through sperm count.
2. When did you perform the questionnaire?
3. In Results, you have two groups, however in Material and method you did not say how you allocated the subjects
4. How do you responders know what blood test they underwent and their results? In my opinion is difficult for an average person to know/remember these data...
5. You said that the patient evaluated the Morphology...of what?
6. Group age of 50-56 years is missing
7. Considering that male infertility is a multifactorial condition, how did you assesed the fertility status? Most of your responders are young and nowadays in many european countries many families postpone having children...
8. You should provide in tables not only percentages but the number of patients because they are very confusing. Considering that more that three quarters were aged under 30 years, how come 70% have benign prostatic hyperplasia?
In conclusion, although I think your paper has many strenghts and is interesting, it needs extensive modifications.
Comments on the Quality of English Language
Dear Editor, although I am not a native English speaker, I believe that the text needs editing for a better soundness.
Author Response
Reviewer #2
Comments and Suggestions for Authors
Dear authors, after reading your manuscript, I believe it has many drawbacks:
Response: We would like to thank Reviewer #2 for taking the time to review our manuscript and for pointing out all the drawbacks. We are very grateful for all valuable comments and suggestions that helped us improve the quality of the manuscript. Below we respond point by point to the Reviewer's comments. Relevant corrections are highlighted in the revised manuscript bold and with track changes made of Word.
- Firstly, the title is not appropriate, you did not actually prove that illegal use of testosterone affected your subjects fertility status. It is true that testosterone use is linked to male infertility, you have not evaluated the fertility status through sperm count.
Response: We thank Reviewer #2 for this comment. Indeed, the title did not accurately reflect the results of the study and we are very grateful for pointing this out. The title has been changed as below (page 1, lines 2-3):
“Illegal use of testosterone and other anabolic-androgenic steroids in the population of amateur athletes in WrocÅ‚aw, Poland – an unfavorable lifestyle trend in the population of men of reproductive age.”
- When did you perform the questionnaire?
Response: We would like to thank Reviewer #2 for pointing out that the description of the method does not provide detailed information on how the survey was conducted. It was included in the study design section as below (page 4, lines 233-235):
„Recruitment of respondents.
Respondents were recruited by posting a survey on the Facebook pages of public gyms between June 2022 and June 2023 ”
- In Results, you have two groups, however in Material and method you did not say how you allocated the subjects
Response: We thank Reviewer #2 for pointing this out. Missing information about how the subjects were allocated were added to the text on page 4 lines 249-251.
„The way respondents were included in the control and research groups.
The control group consisted of respondents who declared in the survey that they did not use testosterone, and the research group consisted of people who declared that they used testosterone.”
- How do you responders know what blood test they underwent and their results? In my opinion is difficult for an average person to know/remember these data...
Response: We would like to thank Reviewer #2 for taking up this topic. Indeed, it was also surprising for the authors of the survey that so many people were able to indicate type of ;aboratory tests they performed. The tests most frequently performed by respondents included blood count, liver function, TSH, estardiol and total and free testosterone concentrations. The use of testosterone in supra-clinical doses as well as other anabolic-androgenic steroids causes changes in blood count, liver function, estradiol concentration and may lead to changes in thyroid function. The ability of respondents to indicate specific tests may indicate a high level of respondents' knowledge about the use of these prohibited substances, but it cannot be ruled out that that such a distribution of answers is random. However, extensive knowledge about testosterone and other anabolic-androgenic steroids seems to confirm the data on other substances used simultaneously and the purpose of their use. This was already reflected in the discussion previously, but Reviewer #2's question prompted us to expand it as follows (page 19 , lines 274-281):
„Surprisingly, most of them pointed to specific laboratory tests they had performed in connection with the use of testosterone and other anabolic-androgenic steroids despite their lack of medical education. The most common tests included blood count, estradiol, TSH, total and free testosterone, and liver function. The fact that they were able to indicate specific types of laboratory tests and such a distribution of answers may also indicate that they have relatively broad knowledge about the use of testosterone and other anabolic-androgenic steroids but it cannot be ruled out that that such a distribution of answers is random.“
- You said that the patient evaluated the Morphology...of what?
Response: We would like to thank Reviewer #2 for pointing out the use of the wrong word. The morphology has been corrected to blood count ( page 11, Table 2)
- Group age of 50-56 years is missing
Response: As suggested by Reviewer #2, the missing age group has been added. Additionally, in table 1, zeros in front of numbers have been removed in some places and the text has been linguistically and grammatically corrected (page 6, Table 1)
- Considering that male infertility is a multifactorial condition, how did you assesed the fertility status? Most of your responders are young and nowadays in many european countries many families postpone having children...
Response: We thank Reviewer #2 for this comment. As Reviewer #2 noted earlier, we did not determine male fertility based on semen analysis. The study took the form of an anonymous survey that was intended to answer the question of the scale of use of illegal testosterone and the profile of people using it. The form of the survey does not allow for fertility assessment. We can only assess factors that may potentially affect fertility. We agree with Reviewer #2 that due to the frequent postponement of the decision to become a parent until a later period in life, the phenomenon we have demonstrated may not have a significant impact on fertility at a later age and requires further research. This was already reflected in the section on the limitations of our work and has now been expanded (page 19, lines 801-807) and, as previously suggested, in the title change (page 1, lines 2-3).
“The survey form we have used allowes only for the estimation of behaviors that may potentially reduce fertility, but not the reduction in fertility itself.”
„Although the profile of a man using testosterone that we established corresponded to people with the best socio-demographic conditions for having children, most of our respondents were young and currently more and more people are postponing the decision to have children until later in life. Whether the disturbing trend we have demonstrated will translate into fertility reduction requires further investigation“
- You should provide in tables not only percentages but the number of patients because they are very confusing.
Response: Following Reviewer‘s #2 suggestion, changes have been made to the tables:
- table 2: new values ​​in the format % (k), where k - number of answers to a given question (description in the footer) (pages8-11, lines 466-471)
- table 3: new values ​​in the format: % (k), where k - number of answers to a given question (description in the footer) (pages 13-14, lines 548-551).
Additionally, the missing exponential notation has been supplemented in Table 4 and some levels have been marked in bold (pages 14-16, lines 573-582)
The text in all tables has been linguistically and grammatically corrected.
- Considering that more that three quarters were aged under 30 years, how come 70% have benign prostatic hyperplasia?
Response: In response to Reviewer #2's question, I would like to clarify that the question concerned the respondents' knowledge of the possible side effects of testosterone and other anabolic-androgenic steroids, and not the occurrence of these complications. Our use of: "Side effects of the use of testosterone and other anabolic-androgenic steroids known by the respondents" could be misleading and has therefore been replaced by: "Awareness of possible side effects of the use of testosterone and other anabolic- androgenic steroids" (page 11, Table 2)
„Awareness of possible side effects of the use of testosterone and other anabol-ic-androgenic steroids“
In conclusion, although I think your paper has many strenghts and is interesting, it needs extensive modifications.
Response: We would like to thank Reviewer #2 for noticing the strengths in our work, pointing out weaknesses and necessary corrections, and also for allowing us to improve the manuscript. All suggested corrections have been made and we hope that the work in this form will be suitable for publication.
Comments on the Quality of English Language
Dear Editor, although I am not a native English speaker, I believe that the text needs editing for a better soundness.
Response: The paper has been carefully revised by a specialist in English linguistics to improve the grammar and readability. The corrections are highlighted in the revised manuscript with bold and track changes made of Word.

Round 2
Reviewer 1 Report
Comments and Suggestions for Authors
The authors addressed my comments properly. However, the overall quality of the manuscript is intermediate to low due to its nature.
Reviewer 2 Report
Comments and Suggestions for Authors
Dear Authors, I have after rereading the article, I noticed that you made the necessary changes. From my point of view, the manuscript deserves to be published